# Conjugation Mechanism for Pneumococcal Glycoconjugate Vaccines: Classic and Emerging Methods

**DOI:** 10.3390/bioengineering9120774

**Published:** 2022-12-06

**Authors:** Victor Morais, Norma Suarez

**Affiliations:** Departamento de Desarrollo Biotecnológico y Producción, Instituto de Higiene, Facultad de Medicina, Universidad de la República, Montevideo 11600, Uruguay

**Keywords:** pneumococcus vaccine, glycoconjugate vaccines, protein carbohydrate conjugation, reductive amination, CDAP

## Abstract

Licensed glycoconjugate vaccines are generally prepared using native or sized polysaccharides coupled to a carrier protein through random linkages along the polysaccharide chain. These polysaccharides must be chemically modified before covalent linking to a carrier protein in order to obtain a more defined polysaccharide structure that leads to a more rational design and safer vaccines. There are classic and new methods for site-selective glycopolysaccharide conjugation, either chemical or enzymatic modification of the polysaccharide length or of specific amino acid residues of the protein carrier. Here, we discuss the state of the art and the advancement of conjugation of *S. pneumoniae* glycoconjugate vaccines based on *pneumococcal* capsular polysaccharides to improve existing vaccines.

## 1. Introduction

*Streptococcus pneumoniae* are encapsulated Gram-positive bacteria that were first isolated independently by Louis Pasteur and George Sternbern in 1880 [1,2]. This bacterium is an important human pathogen that causes pneumonia, otitis media, meningitis, and bacteremia. It is a major global cause of morbidity and mortality, especially in children, the elderly, and immuno-compromised populations, with approximately a million deaths yearly worldwide [3,4].

One of their main virulence factors is the capsular polysaccharide (CPS), which is involved in mediating direct interactions between the bacteria and its environment, protecting the pathogen from the host immune response mechanisms [5,6]. Based on their CPS, they have been classified into more than 90 different specific types according to the Danish classification system [7,8]. Recently, serotype number 100 was identified [9].

Furthermore, this pathogen mainly infects humans, first colonizing the nasopharynx, which is its only reservoir in nature. Most clinically isolated *S*. *pneumoniae* have been susceptible to penicillin, to the β-lactams group in general, and to other antimicrobial drugs. However, due to the global emergence of resistant and multidrug-resistant strains and the fact that, in many cases, the disease can progress quickly before introduction of antimicrobial treatment, pneumococcal disease is a challenge to treat [10,11].

The circulation of many different serotypes constitutes a challenge in the development of an effective vaccine due to the limited cross reactivity between each type [12,13,14]. Consequently, optimal vaccines should include the most prevalent and virulent serotypes. Pneumo-Bacterin, a serotype-specific whole-cell heat-treated vaccine, was the first vaccine against pneumococcus, recorded in 1909 in the USA [1]. The identification of pneumococcal bacteria serotypes and their establishment as critical virulence factors by Dochez and Avery in 1917 was the starting point for the use of pneumococcal polysaccharide vaccines, which are still currently used [1,15,16,17,18].

In 1929, Avery conjugated (covalently coupled) pneumococcal polysaccharides to proteins to improve immunogenicity. They used horse globulin and egg albumin as a carrier protein, and the CPS was presumably serotypes 1 and 2 [1,19]. This improvement accomplishes real clinical value many years later with the development of conjugate pneumococcal vaccines [6,14,20]. Meanwhile, only unconjugated polysaccharide vaccines were used.

Polysaccharide pneumococcal vaccines can stimulate antibody production by B cells in the absence of T helper cells. This type of response is termed thymus independent respond or simply T independent response [21]. Most of the polysaccharides cannot be processed and presented in association with Major Histocompatibility Complex (MHC) and therefore cannot be recognized by helper T cells (an exception occurs with some zwitterionic polysaccharides like *S. pneumoniae* CPS 1). In general, polysaccharide antigens are composed of repeated units that induce cross linking of B cell receptors (BCR), leading to the activation of the B cell. Due to the lack of T helper collaboration, antibodies are generally of low affinity and are composed mainly of IgM with limited isotype switching to IgG and IgA, and the production of memory B cells is also diminished. Consequently, T independent antigens are poor immunogens, particularly in young infants [21,22].

The need for more effective vaccines against these capsular polysaccharides led to their conjugation to immunogenic carrier proteins, resulting in conjugates that evoked a better T-cell dependent memory response. These vaccines can be processed by antigen presenting cells (APCs) and peptides from the protein component can be presented in association with the MHC and be recognized by helper T cells. The activation of T helper cells induces a better response, including antibody affinity maturation, isotype switching, and the induction of better secondary responses by B memory cells [22].

In 1980, Schneerson and coworkers, taking into account Avery’s work, developed a Haemophilus influenzae type b (Hib) conjugated vaccine using polyribosylribitol phosphate polysaccharide (PRP) conjugated to CRM-197 (a non-toxic recombinant variant of diphtheria toxin) [1,23,24]. Some years later, due to the success of this vaccine, Wyeth pharmaceuticals conjugated the CPS of the seven most common types of Streptococcus pneumoniae to CRM197, generating the first conjugated vaccine against pneumococcus (Prevnar7^®^) [23,25].

The introduction of conjugate pneumococcal vaccines (PCVs) in the 21st century has been widely accepted, and they are used in children in most countries. The currently approved and in-use PCVs are shown in Table 1.

Prevnar vaccines uses CRM197 as the carrier protein, while Synflorix vaccine uses three types of protein: diphtheria toxoid (DT) for CPS type 19F, tetanus toxoid (TT) for CPS type 18C, and non-typeable *Haemophilus influenzae* protein D (NTHI PD) for the other eight CPS types found in the vaccine [28]. Although vaccination with PCVs includes most of the prevalent serotypes of pneumococcus, the World Health Organization (WHO) reported the emergence of non-vaccine serotype circulation and serotype replacement, leading to an increased proportion of pneumococcal disease of these serotypes in both children and adults [29,30].

In addition to the CPS and carrier protein, many other factors can affect the effectiveness of PCVs as a method of conjugation, including the chemical characteristics of the particular conjugate, the presence or absence of either polysaccharide protein cross-linking or spacer arms, and the size of the conjugated polysaccharide hapten, among others.

Purified polysaccharides can be used in their native form or size-reduced through mechanical or chemical methods. Microfluidization, acidic hydrolysis, or hydrogen peroxide treatments are the most common. These treatments have implications in the protein conjugation step. The polysaccharide size reduction reduces the viscosity of the solution, facilitating the chemical coupling to the protein carrier. Moreover, as an alternative to CPS purification, chemical synthesis or *E. coli* glycoengineering could be used [27].

Notably, the conjugation methods are very important in the development of an effective vaccine, and new methods have recently emerged to obtain better immune responses and more defined products.

## 2. Classic Conjugation Methods

The importance of conjugating sugars to a protein was reported by Avery and Goebel about a century ago to enhance antigenicity to the sugars [19,20].

Despite the high content of OH groups in polysaccharides, their reactivity is insufficient to be directly bound to proteins. Therefore, they require activation before being covalently coupled to a protein. The optimal activation of a polysaccharide depends on its structure [31]. It is possible to introduce new functional groups into the structure of these polysaccharides through chemical or enzymatic reactions that can give them novel physical–chemical and biological properties, which can be used for further covalent coupling to proteins.

In this sense, a variety of conjugation chemistry methodologies for the activation of polysaccharides have been used to couple carbohydrate antigens to protein carriers. Among them are the traditional and widely used reductive amination and activation by cyanogen bromide or related groups as 1-Cyano-4-Dimethylaminopyridine Tetrafluoroborate (CDAP), leading mainly to the formation of high molecular weight, heterogeneous, and relatively undefined network-like structures [32,33,34,35].

### 2.1. Reductive Amination

This method has been used in most of the licensed PCVs and consists of sugar–protein conjugation through Schiff base formation between sugar-derived aldehydes and amines in proteins followed by a reduction [36,37]. The first step is the introduction of aldehyde groups along the polysaccharide chain by partial oxidation with sodium periodate. Cis-diols in the sugar ring and sialic acid residues are the most sensitives zones to generate aldehydes by periodate oxidation [27]. The carrier protein is then added to the reaction. Aldehydes from sugars are typically reacted with the lysine residues of the carrier protein. This methodology consists of the initial formation of bonds by Schiff bases between a carbonyl group in the carbohydrate and an amino group in the protein (Figure 1). The Schiff base is then specifically reduced with the weak reductant sodium cyanoborohydride to a more stable amine [27,33].

Many variations have been employed to improve the conjugation process. Linkers can be used to insert chemical handles for conjugation to reduce steric problems. Another alternative is the use of ammonium salts to provide a reactive amine ready for coupling or the use of dihydrazide spacers. Inserted amines can be reacted directly with carboxylic groups of protein acid residues or coupled to a variety of bifunctional linkers [27].

Reductive amination has been widely used in glycoconjugate synthesis because of its simplicity, but it has important drawbacks, including a low yield, low coupling efficiency, and the risk of the partial degradation of the carbohydrate structure.

The conjugation method can influence the immunogenicity and the functionality of the antibodies induced, leading to deficient cross-reactivity with wildtype serotypes. This was demonstrated for serotype 19F and 19A when their conjugation by reductive amination and cyanylation was compared [38]. These studies showed that the conjugation of the 19F polysaccharide using reductive amination induced a new epitope that was not present in the native form of the 19F polysaccharide or using cyanylation as a conjugation method [38].

In addition, Poolman and colleagues reported that the periodate oxidation step before reductive amination could open the saccharide ring structure, which may lead to ring-opened conjugates, depending on the serotype. The authors proposed that the ring-opened polysaccharides may reclose after the conjugation process, in some cases, causing the polysaccharides to form additional new conformations, resulting in new epitopes [38].

Besides, reductive amination is often attained to a certain extent under basic conditions, beginning with the work of Jennings and Lugowski in 1981 [39]. They used these conditions because the polysialic acids in meningococcal B and C capsular polysaccharides are acid labile. To avoid possible degradation, they proposed carrying out the conjugation between the terminally oxidized polysialic acid and tetanus toxoid in phosphate buffer at pH 9.0 in the presence of sodium cyanoborohydride [39,40]. From this point, the reductive amination standard procedure was carried out in basic conditions, although Schiff’s base can be formed under either basic or acidic conditions [40]. However, Zou and coworkers observed significant degradation of the polysaccharide under basic conditions during conjugation of several pneumococcus capsular polysaccharides by reductive amination. The authors found that performing the conjugation of oxidized polysaccharides to bovine serum albumin (BSA) in slightly acidic media improved the reductive amination and prevented degradation, particularly with β-elimination-susceptible polysaccharides [40].

### 2.2. 1-Cyano-4-dimethylaminopyridine Tetrafluoroborate (CDAP) Cyanilation

This methodology comes from another classic strategy for the conjugation of carbohydrates and proteins, consisting of the activation of polysaccharides with cyanogen bromide (CNBr), as described by Kohn and Wilchek in 1984 [35].

The polysaccharide aliphatic hydroxyl groups are known not to be sufficiently nucleophilic to react directly with CNBr, and it is not possible to carry out the reaction at neutral pH. These conventional procedures are characterized by the use of CNBr in a strongly basic reaction medium. The activation yields are only 0.5–2%, which means the use of large amounts of CNBr, with all the concomitant health risks. Another disadvantage of these processes is that the activated polysaccharide contains both cyanate ester and imido carbonate structures in different proportions. The cyanate ester groups are hydrolyzed at a high pH, while the imido carbonates form inactive carbonates on hydrolysis at a more acidic pH.

Although CNBr is a low-cost reagent, it is very toxic and its use requires special attention. In addition, sensitive polysaccharides can be damaged by the high pH conditions required for CNBr activation [41].

Despite these disadvantages, the conventional procedure is one of the most popular methods used for the activation of polysaccharide-type matrices for protein immobilization and in affinity chromatography. As an alternative strategy, the electrophilicity of CNBr can be increased by means of a suitable “cyano-transfer” agent, which does not require the presence of a strong inorganic base in the reaction medium and thus makes it possible to avoid the base-dependent side reactions described above. These complexes are more electrophilic than CNBr and thus capable of cyanylating hydroxyl groups in their protonated form in the polysaccharide to create cyanate esters at a lower pH than in the conventional process. One of the classical cyanotransfer procedures is based on the formation of a highly reactive salt-type complex between CNBr and a cheap and easily accessible tertiary amine, TEA (CTEA). The yield of the reaction was highly improved, but CTEA is unstable and decays at temperatures above −10 °C [35,42]. In the same way, the authors found that 1-cyano-4-(dimethylamino) pyridiniumtetrafluoroborate (CDAPBF_4_ or CDAP) was a highly efficient activating agent for polysaccharide resins [42].

Indeed, Lees and coworkers introduced CDAP for protein–polysaccharide conjugate vaccines, using it as a cyanylating agent for the activation of soluble polysaccharides such as pneumococcal type 14 polysaccharide and others [43,44]. CDAP, which is easier to use compared to CNBr, has the advantage that it can activate polysaccharides at a lower pH than CNBr and with fewer side reactions. Unlike CNBr, CDAP-activated polysaccharides can be directly conjugated to proteins, creating a simpler process (Figure 1). Moreover, CDAP-activated polysaccharides can be functionalized with a diamine or a dihydrazide to make amino- or hydrazide-derivatized polysaccharides. Further optimization of CDAP polysaccharide activation with pH control and buffer optimization was later described [41,45].

Lees and colleagues reported some important concerns to achieve an optimum activation of the polysaccharide: the stability of CDAP, the reaction with the polysaccharide hydroxyls, and the stability of the activated polysaccharide. They demonstrated that the CDAP reaction with the polysaccharide requires for optimization the increase of the pH and performing the reaction in the cold. These conditions reduce CDAP hydrolysis, in-creasing the active reagent available for reaction, and have minimal effect on the rate of activation. The authors conclude that to increase the efficiency of activation of polysaccharides with CDAP requires a balance between these concerns [41].

Currently, the technique has widespread use in both research and licensed vaccines, but studies on optimization of CDAP chemistry remain scarce [41].

## 3. Emerging Methods

Currently, in order to overcome the drawbacks of the classic methodologies, some new conjugation techniques have been applied to improve conjugation of pneumococcus capsular polysaccharides. Some of these are in the lab phase, but others are in clinical assays [46,47,48].

### 3.1. The Galactose Oxidase (GO) Method for Reductive Amination

Classic reductive amination needs a previous step using periodate-controlled oxidation of the sialic acid side chains and then subsequent coupling to protein carriers [46]. The oxidation of the diol systems can provoke structural modifications of the polysaccharide antigens and can also result in a degradation phenomenon, causing a reduced antibody cross-reactivity with the native polysaccharide [46,47,48]. In a recent publication, Duke and co-workers proposed an alternative to periodate oxidation prior to reductive amination. They showed that an enzyme isolated from the fungus *Fusarium sp*., galactose oxidase (GOase), can generate aldehyde groups in a defined position that are suitable to prepare a conjugate vaccine against *S. pneumoniae* [49]. In a compatible polysaccharide, the GOase enzyme generates a site-specific aldehyde motif at position C6 of the galactose suitable for conjugation. Furthermore, the aldehyde group can be turned to the original primary alcohol group by reduction with sodium borohydride.

They worked with the CPS *of S. pneumoniae* serotype 14 (Pn14p) and confirmed that the GOase generated aldehyde motifs in a site-specific and reversible fashion. Direct comparison of the Pn14p derivatized by either GOase or NaIO_4_ indicated that the GOase treatment resulted in a minimal decrease in the structural integrity of the CPS. Immunization by the conjugate with CRM_197_ synthesized using GOase and reductive amination provided a better humoral response over the periodate-oxidized group. Furthermore, the conjugate vaccine obtained better functional protection measured by opsonophagocytic killing and a lethality challenge in mice [49].

### 3.2. Multiple Antigen Presenting System (MAPS) Platform

MAPS is a platform that aims to emulate the antigenic and immunologic strengths of whole-cell vaccines but in a precise acellular system. MAPS uses purified polysaccharides and proteins as immunogens providing a defined composition [50]. Distinctive from other approaches, the antigen components in the MAPS system are specifically reassembled into a defined macromolecular complex, based on the idea that mimicking some chemical and physical features, as with virus-like particles and native viruses, could provide the multipronged protection characteristic of whole-cell vaccines [50]. The MAPS platform utilizes the high affinity noncovalent interaction between biotin and rhizavidin, a biotin-binding protein from *Rhizobium etli*, and uses it to create a supramolecular structure of proteins and carbohydrates [50,51].

Rhizavidin is modified and recombinantly expressed in *Escherichia coli*, fused to the carrier protein. The target polysaccharide is biotinylated by the CDAP method, providing a scaffolding matrix for the complex [52]. After mixing and incubation, the carrier protein attaches to the polysaccharide backbone via a strong affinity linkage of biotin and rhizavidin, resulting in a macromolecular construct. The resulting assembled MAPS complexes can be easily isolated from unbound antigens by size-exclusion chromatography. The association between polysaccharide and protein antigens is very stable and resistant to spontaneous dissociation.

Using this platform, ASP3772, a novel 24-valent pneumococcal vaccine of Affinivax labs (GSK), has recently concluded clinical phase 2 with promising results [51]. ASP3772 contains 24 polysaccharides, including the 13 serotypes contained in PCV13 and PPSV23 (unconjugated polysaccharide vaccine) and 10 contained in PPSV23 exclusively. The ASP3772-included serotypes are: 1, 3, 4, 5, 6A, 6B,7F, 9V, 14, 18C, 19A, 19F, and 23F (shared with PCV13 and PPSV23, except 6A); 2, 8, 9N,10A, 11A, 12F, 15B, 17F, 22F, and 33F (only shared with PPSV23); and the 20B serotype not presented in any vaccines.

Serotypes are individually biotinylated and complexed with a fusion protein consisting of rhizavidin fused to two pneumococcal protein segments derived from genetically conserved surface protein genes (sp1500 and sp0785) [52].

The vaccine was studied in two different age cohorts (18–64 and 65–85 years of age), and it was well tolerated. The most frequently reported systemic reactions were headache, myalgia, and fatigue. Self-limited tenderness and pain at all dose levels were the most frequently reported local reactions. Similar results were obtained with the administration of PCV13 as the control [51].

Robust opsonophagocytic activity responses for all serotypes were observed for all ASP3772 dose groups in both age cohorts. Older adults (aged 65–85 years) who received ASP3772 showed functional antibody responses comparable to or higher than those shared with PCV13 and PPSV23 [51]. Furthermore, investigators found that the multimolecular complex enhances the generation of specific B-cell responses to PSs and the Th1/Th17 immune responses to proteins [47,48]. It has been established that CD4+ T helper type 17 (Th17) cell response leads to the reduction of pneumococcal colonization, killing and clearing of pneumococci by recruiting neutrophils to the site of infection. This type of response could reduce pneumococcal colonization, promoting clearance and is independent of CPS type [53,54,55,56,57].

### 3.3. Protein Glycan Coupling Technology (PGCT)

Protein Glycan Coupling Technology (PGCT) uses purposefully modified bacterial cells to produce recombinant glycoconjugate vaccines in one step (Figure 2). The protein–carbohydrate conjugation comes from a protein N-glycosylation system of *Campylobacter jejuni*. The machinery required for glycosylation is encoded by 12 genes clustered in the pgl locus [58,59]. A primary consensus sequence must be added to achieve N-glycosylation. The N terminus, extended to D/E-Y-N-X-S/T, and a negatively charged side chain at position-2 are necessary for recognition by the bacterial oligosaccharyl transferase PglB [59].

The PGCT comprises three components cloned in most cases in *Escherichia coli* [60,61,62]. The first component is to clone and express the glycan of interest (in the case of *S. pneumoniae*, capsular polysaccharide). The correct expression of the polysaccharide can be difficult. In many cases, a defined gene cluster has all the requisite genetic information and can be transferred properly to *E. coli* for expression. The glycan locus is usually introduced on a low copy plasmid. However, in a more complex genetic background, it may be more convenient to use the bacteria from which the glycan originated as a host cell [61].

The second component is the carrier protein design and expression. It is necessary to include the extended glycosylation sequon (D/EXNYS/T) for PglB in the construction. The use of the flexible region of the target protein results in more efficient glycosylation by PglB; some studies have also indicated that multiple sequons can be added for the production of a more heavily glycosylated vaccine [61].

The final component is the glycosylation component from *C. jejuni*. Since the original identification of PglB, many orthologues have been identified, but the original component is still commonly used. For example, new PglBs have been identified from two Desulfovibrio species. These PglBs do not require the negatively charged amino acid at position-2 and are able to glycosylate a shorter N-X-S/T sequon [61,63].

**Figure 2 bioengineering-09-00774-f002:**
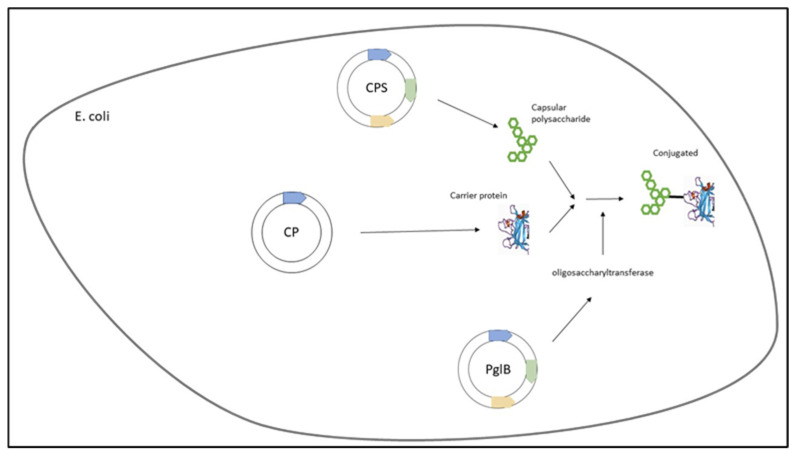
PGCT technology for the production of glycoconjugates. CPS, Pneumococcus capsule. CP carrier protein. PglB, glycosylation component from *C. jejuni.* Recombinant proteins are inserted into an inducible plasmid from the pEXT family. Conjugation was obtained by bacterial mechanical lysis and Ni affinity chromatography purification [64].

Using this methodology, Reglinski and coworkers created a recombinant PCV containing serotype 4 CPS linked to the *S. pneumoniae* proteins NanA, PiuA, and Sp0148 [64]. They also found that the introduction of the *Campylobacter jejuni* UDP-glucose 4-epimerase gene GalE (gne) into *E. coli* is necessary to improve the yield of the resulting glycoprotein [64]. According to the authors, the main advantages of this method included the low cost of the recombinant approach and the easy purification process with a single step Ni^2+^ affinity chromatography procedure that can readily be scaled up for manufacture.

The results showed a strong antibody response in mice to both the capsule and the carrier protein antigens. The CPS4-PiuA glycoconjugate induced similar anti-capsular antibody responses to the commercial PCV Prevnar-13. The antibody responses to the PGCT glycoconjugates effectively opsonized *S. pneumoniae* serotype 4 and promoted neutrophil phagocytosis as effectively as the antibodies generated by vaccination with PCV13. In the challenge assays, the authors found that vaccination with the PGCT glycoconjugates protected mice against meningitis and septicemia with the same efficacy as vaccination with PCV13. In other studies, researchers have demonstrated that a vaccine created by PGCT was as effective as Prevnar-13 [64].

## 4. Conclusions

PCV vaccines have become a key tool for pneumococcus prevention in recent decades. Several conjugation methods have been used and licensed in PCVs. All commercial vaccines use only two methods: reductive amination and CDAP conjugation.

Despite, the significant success of these vaccines, there are important drawbacks of these methods, including low yield and component degradation. Consequently, different novel designs as MAPS, PGCT, and GOase have recently emerged (Table 2). In the case of the GOase method, an attempt was made to solve some of the drawbacks of reductive amination, such as the possible degradation and the low defined coupling, characteristics of the traditional methods. On the other hand, MAPS technology seeks a new type of conjugated vaccine that includes better antibody responses and generation of Th17 response to obtain extended protection, regardless of the CPS type. Finally, The PGCT technology proposes the recombinant production of all components, achieving an easy production process and a methodology easily adaptable to further improvements in the selection of carrier proteins and polysaccharides. Currently MAPS, PGCT, and GO methods are still under development, but it is likely that they will become part of the new generation of PCVs.

## Figures and Tables

**Figure 1 bioengineering-09-00774-f001:**
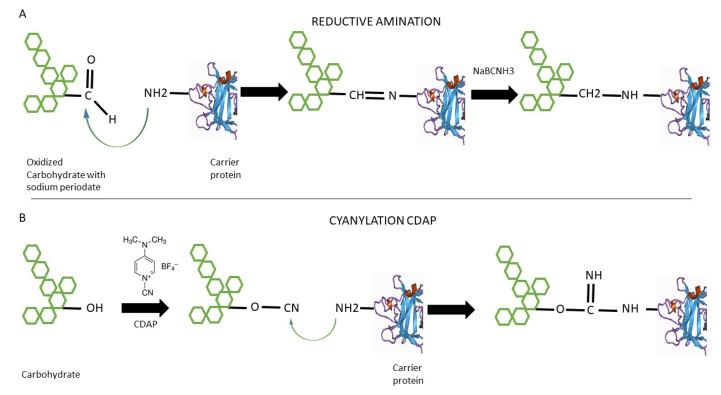
Conjugation by Reductive Amination and CDAP.

**Table 1 bioengineering-09-00774-t001:** Currently used PCVs approved by FDA and/or EMA. * Displaced by PCV13. Abbreviations: TT, tetanus toxoid; CRM197, cross-reactive material 197; NTHI PD, non-typeable *Haemophilus influenzae* protein D; DT, diphtheria toxoid; FDA, Food and Drug Administration; EMA, European Medicines Agency; GSK GlaxoSmithKline. References [26,27,28].

Vaccine	Producer	CPS Type	Protein	Conjugation Method
Prevnar PCV7 *	Pfizer	4, 6B, 9V, 14,18C, 19F, 23F	CRM197	Reductive amination
Prevnar PCV13	Pfizer	1, 3, 4, 5, 6A, 6B, 7F, 9V, 14, 18C, 19A, 19F, 23F	CRM197	Reductive amination
Prevnar PCV20	Pfizer	1, 3, 4, 5, 6A, 6B, 7F, 8, 9 V, 10A, 11A, 12F, 14, 15B, 18C, 19A, 19F, 22F, 23F, 33F	CRM197	Reductive amination
Synflorix PCV10	GSK	1, 4, 5, 6B, 7F, 9 V, 14, 18C, 19F, 23F	NTHi PD, DT, TT	CDAP
Vaxneuvance PCV15	Merck	1, 3, 4, 5, 6A, 6B, 7F, 9 V, 14, 18C, 19F, 19A, 22F, 23F, 33F	CRM197	Reductive amination

**Table 2 bioengineering-09-00774-t002:** Main advantages and disadvantages of currently conjugation techniques used in pneumococcal vaccines.

Method	Main Advantages	Main Disadvantages
Reductive amination	Proven efficacy in human use in commercial vaccines	Low yield Component degradation
CDAP	Proven effectivity in human use in commercial vaccines	Low yield
Goase	Improved yield and reduced component degradation	Not proven in humans. Still in preclinical phase
MAPS	Improved immunogenicity	Recently started in clinical phase
PGCT	Improves and facilitates vaccine production	Not proven in humans. Still in preclinical phase

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
