# Peer review of "Conjugation Mechanism for Pneumococcal Glycoconjugate Vaccines: Classic and Emerging Methods"

_bioengineering, 2022, doi:10.3390/bioengineering9120774_

Round 1

Reviewer 1 Report

The authors present a review that focuses on the different conjugation methods of peptides to the pneumococcal polysaccharides. Although the study is interesting, it is very limited in addressing all the complexity behind the development of the glycoconjugate vaccines against different Streptococcus pneumoniae serotypes.

Some main concerns in the manuscript are:

Activation of the adaptive immune response to this vaccine is poorly described. In particular, the maturation of B cells and the production of IgM or IgG is not addressed, being pivotal in explaining the need for conjugating peptides to the polysaccharides. Authors should address the consequences of conjugating

The authors don´t address the strategies used for the extraction and production of the glycan component that must be modified later in order to conjugate peptides. Is the purification strategy determining to later perform some chemical reactions? could it be more appropriate than others? Is it known whether the characteristics of polysaccharides from different serotypes determine the conjugation strategy?

Finally, the authors present a brief conclusion paragraph but do not discuss the challenges of this kind of vaccine and whether the conjugation reaction can be improved.

Minor comments.

1.     Introduction, lines 22, 23. Specify the other infectious diseases.

2.     Line 25: Million of deaths in what year? Where?

3.     Line 33. Do the authors mean Beta-lactam antibiotics?

4.     Line 34. Resistant strains to beta-lactams or multidrug-resistant. Expand and specify.

5.     Two paragraphs in the classic conjugation methods, lines 70 to 83. Remove.

6.     When authors talk about the different vaccines containing different serotypes, are all serotypes conjugated the same?

7.     Line 226, how the polysaccharide is biotinylated?

8.     It would be interesting to have a figure/picture showing the PGCT process in point 3.3.

Reviewer 2 Report

The topic of conjugation methods for pneumococcal glycoconjugate vaccines the authors proposed is interesting. However, the main section on conjugated methods lacks focus and is a compilation of literature sources; several paragraphs are the way to general and do not really provide meaningful up-to-date information and reasonable discussion. I suggest the review should critically evaluate the available volume of information.

1.     The logic of the review should be improved, such as in the introduction section, with the first three paragraphs starting with “S. pneumoniae,” accumulating the related content. It is better to provide a better manuscript flow, allowing readers to follow your stream.

2.     In the introduction section, “The circulation of many different serotypes constitutes a challenge in the development of an effective vaccine,” where one reference should be included. By the way, what is the challenge, and how will it affect the development of vaccines?

3.     In the manuscript, short forms should be defined when shown for the first time. One example is the CDAP; the author should define it as 1-Cyano-4-Dimethylaminopyridine Tetrafluoroborate when it first comes out. Others should be carefully checked and revised in the whole manuscript.

4.     The main section on conjugated methods lacks focus and is a compilation of literature sources; several paragraphs are the way to general and do not really provide meaningful up-to-date information and reasonable discussion. I suggest the review should critically evaluate the available volume of information.

5.     Where in the manuscript, more efforts should be brought to arrange and organize extra figures to describe the methods with a comparison to reflect the strengths of each, which might be in a table format.

Reviewer 3 Report

The introduction of glycoconjugate pneumococcal vaccines has resulted in a dramatic reduction in pneumococcal disease in North America and Europe. While the vaccines are effective the methods of manufacturing the vaccines are not without challenges.  The authors have written an interesting but brief review of methods used to prepare pneumococcal vaccines. Overall, the manuscript should be edited to expand on some of the excellent points that are raised. In addition, the organization of the section on classical methods needs to be improved for readability.  My specific comments are listed below.

Lines 45 -53 – The observations of Avery that conjugation improved the immune response had a significant impact on the development and subsequent licensure of glycoconjugate vaccines prior to the 21st century.  This section should be revised with references, to acknowledge development and licensure of conjugate vaccines prior to the introduction of the first pneumococcal conjugate.

Line 63-68 – The authors raise some very important points regarding the importance of conjugation method to the effectiveness of the vaccine.  The paragraph should be expanded to more fully describe the heterogeneity of conjugation and variations observed with immune responses.

Lines 70 -84 – The initial paragraphs under the heading “Classical Method” seems out of place and confusing. It seems to be describing problem with the way methods are published in the literature rather than the methods themselves. Perhaps the ideas expressed should organized in a separate section relating to currently published information,

Line 295-302 – The authors have written an interesting but brief review of methods for preparing pneumococcal conjugate vaccines. The “Conclusions” as written are too brief to impart very much information to the reader about what was described above and where this can lead in vaccine development.

Lines 308-408 – The formatting of the references is inconsistent, and the fonts used are not uniform.

Round 2

Reviewer 1 Report

Thank you for the new version. It has improved the quality of the manuscript.

Some suggestions:

Reconsider the order of some sections/paragraphs.

Line 31, Authors may consider saying (...) "this pathogen mainly infects humans". report of this bacteria infecting dogs for example are available.

Line 143: E. coli must be in italics.

Line 289: Fusarium sp., must be in italics.

Line 421: Figure 2 legend needs a better description. Plasmids are poorly described as well as the expression method. Are polysaccharides retained inside the bacteria or secreted? are the plasmid promoters constitutive or inducible?

Author Response

We want to thank the reviewer for his valuable comments.

All suggestions were taking into account. Lines 31, 143, 289 were corrected.

Figure 2 legend was better described. 

Best regards,

Reviewer 2 Report

It seems the manuscript has to be revised further according to the previous suggestions. And there is no indication that the revision has been done in the new version. What I can see is the update of the references.

Author Response

Dear Reviewer,

We want to thank the reviewer for his valuable comments. In the last version something occurred with track changes and some Word version could not showed it.

We hope that in this new version old and new changes can be showed. Besides this, we use a red font colour to mark the changes

Best regards,

  1. The logic of the review should be improved, such as in the introduction section, with the first three paragraphs starting with “S. pneumoniae,” accumulating the related content. It is better to provide a better manuscript flow, allowing readers to follow your stream.

This section was Improved and extended.

  1. In the introduction section, “The circulation of many different serotypes constitutes a challenge in the development of an effective vaccine,” where one reference should be included. By the way, what is the challenge, and how will it affect the development of vaccines?

Paragraph was Included with references (line 38)

The circulation of many different serotypes constitutes a challenge in the development of an effective vaccine because the limited cross reactivity between each type [12–14]. Due to this fact, vaccines should include the most prevalent and virulent serotypes.

  1. In the manuscript, short forms should be defined when shown for the first time. One example is the CDAP; the author should define it as 1-Cyano-4-Dimethylaminopyridine Tetrafluoroborate when it first comes out. Others should be carefully checked and revised in the whole manuscript.

Definition was included (line 202)

  1. The main section on conjugated methods lacks focus and is a compilation of literature sources; several paragraphs are the way to general and do not really provide meaningful up-to-date information and reasonable discussion. I suggest the review should critically evaluate the available volume of information.

New information has been included. It is difficult to find the confusing paragraphs to make them better. We tried to describe the existing methods and gave some examples of how they have been used and their advantages and disadvantages. We provide also the latest conjugation methodologies that are described in the literature up-to-date.

  1. Where in the manuscript, more efforts should be brought to arrange and organize extra figures to describe the methods with a comparison to reflect the strengths of each, which might be in a table format.

Figure 2 and table 2 with advantage and disadvantages were included.

Reviewer 3 Report

The authors have addressed my concerns.

Author Response

We want to thank the reviewer for his valuable comments.

Best regards,